# The measurement of mental fatigue following an overnight on-call duty among doctors using electroencephalogram

**Anselm Ting Su**[1]◉*, **Gregory Xavier**[2]◉, **Jew Win Kuan**[3]◉

**1** Department of Community Medicine and Public Health, Universiti Malaysia Sarawak, Kota Samarahan, Malaysia, **2** Kinta District Health Office, Ministry of Health Malaysia, Malaysia, **3** Department of Medicine, Universiti Malaysia Sarawak, Kota Samarahan, Malaysia

◉ These authors contributed equally to this work.
* stanselm@unimas.my

**Data Availability Statement:** All relevant data are within the paper and its Supporting information files.

## Abstract

This study aimed to measure the spectral power differences in the brain rhythms among a group of hospital doctors before and after an overnight on-call duty. Thirty-two healthy doctors who performed regular on-call duty in a tertiary hospital in Sarawak, Malaysia were voluntarily recruited into this study. All participants were interviewed to collect relevant background information, followed by a self-administered questionnaire using Chalder Fatigue Scale and electroencephalogram test before and after an overnight on-call duty. The average overnight sleep duration during the on-call period was 2.2 hours (p<0.001, significantly shorter than usual sleep duration) among the participants. The mean (SD) Chalder Fatigue Scale score of the participants were 10.8 (5.3) before on-call and 18.4 (6.6) after on-call (p-value < 0.001). The theta rhythm showed significant increase in spectral power globally after an overnight on-call duty, especially when measured at eye closure. In contrast, the alpha and beta rhythms showed reduction in spectral power, significantly at temporal region, at eye closure, following an overnight on-call duty. These effects are more statistically significant when we derived the respective relative theta, alpha, and beta values. The finding of this study could be useful for development of electroencephalogram screening tool to detect mental fatigue.

## Introduction

Although sleep deprivation and mental fatigue are two distinguishable conditions [1], they often occurred together in real life, especially among occupations requiring overnight on-call, extended shift, and shift work. Sleep deprivation is a continuous and prolonged period of lack of sleep whereas mental fatigue implies reduced psychophysiological efforts due to prolonged and demanding mental activities [1]. Despite the scientific distinction, both conditions contribute to performance impairments and are often considered interchangeable.

Overnight on-call doctors typically experiencing mental fatigue in addition to sleep deprivation due to the nature of the demanding and busy hospital workload. Overnight on-call causes disturbed sleep [2], and is associated with higher medication errors among doctors [3,

**Funding:** This study was funded by the Brain, Mind and Neuroscience Research Foundation Malaysia (Registration: PPAB-26/2017) (URL: https://www.ypomnm.org) with research grant ID: GL/F05/YPOMNM/2019. The recipient of the grant was AST. The funders had no role in study design, data collection and analysis, decision to publish, or preparation of the manuscript.

**Competing interests:** The authors have declared that no competing interests exist.

4], procedural errors [5], stress reactivity [6], and impairment of attentional networks [7]. Nevertheless, not all post-call doctors are sleepy, mentally fatigue and is at risk of inflicting errors in carrying out their tasks. Hence, from the perspective of occupational health and safety, as well as patient's safety, it is of utmost importance to find a method to detect mental fatigue following sleep deprivation due to overnight on-call duty among doctors, to prevent them from carrying out emergency and surgical procedures which might impose danger to themselves and others.

Electroencephalography (EEG) has been widely used in the evaluation of sleep [8–12], alertness [13–16], and mental fatigue [17–20], especially in the field of transportation medicine [17, 20–24]. There has never been a study to evaluate EEG changes following an overnight on-call duty among the doctors. In this study, we compared the EEG characteristics among the doctors before and after an overnight on-call duty, to identify differences that could be useful for future brain-computer interface system development to detect resting state drowsiness or mental fatigue following overnight on-call duty for healthcare personnel. The objective of this study is to determine the significant differences in the power spectral of EEG rhythms at different brain regions before and after an overnight on-call duty among a group of doctors who actively perform on-call work on regular basis. The findings of this study could contribute to address the gap in the literature by evaluating EEG changes following overnight on-call duty among doctors and develop a brain-computer interface system to detect mental fatigue and drowsiness in healthcare personnel.

## Materials and methods

This was a repeated cross-sectional study on a group of doctors who performed overnight on-call duty in a tertiary government hospital in Sarawak, Malaysia. The inclusion criteria were all doctors who are on a regular on-call work schedule with a working period of more than 12 hours during the overnight calls. We excluded the participants with any history of organic neurophysiological or developmental disorders, neurophysiological injuries, psychiatric illness, taking regular medication, pregnancy at the time of recruitment and illnesses that may affect the neurophysiological function of the brain. All doctors fitting the inclusion and exclusion criteria were invited to participate in the study and the acceptance was voluntary. The participants were free to decide on an on-call day which they would like to carry out this study, but they must ensure that they were present for two sessions, before, and after their on-call work schedule. Prior to the start of the study, all participants would be interviewed to collect the information on their age, gender, education level, duration of work, usual overnight sleep duration, medical history, and alcohol, tea, and smoking consumption. This was followed by the first EEG recording prior to the on-call duty. The participants were then released to perform usual on-call duty and were asked to return on the following day for the second EEG recording. The participants were asked to answer a the Chalder Fatigue Scale [25] questionnaire before and after on-call, prior to the EEG test. The duration of sleep during the overnight on-call duty was determined by self-reporting.

The Chalder Fatigue Scale questionnaire consists of 11 items assessing symptoms of fatigue, such as lack of strength in the muscles, tiredness, lack of energy, sleepiness, memory, and difficulties in concentration, requiring a participant to rate each item on its frequency of occurrence: less than usual, no more than usual, more than usual, and much more than usual. With the use of the Likert scoring system, a score ranging from 0 to 3 is given. The total score of the scale is calculated by adding the rating for each item, which ranges from 0 to 33. With the use of the Likert scale, a score of less than 15 is considered to have 'no fatigue' and 15 and above means 'fatigue' is present. A study by Wessely et al. to distinguish between mild and severe

fatigue among chronic fatigue patients in primary care setting used the similar cut-off point [26]. The internal consistency, calculated by the Cronbach's alpha, for the 11 items of the scale ranged between 0.88 and 0.90 [25].

The EEG recording was performed by a trained and experienced EEG technician who usually performed EEG on patients in the hospital, using Nicolet® Monitor by Natus. The EEG electrodes were arranged according to the International 10–20 systems using the Mitsar® Cap-EEG cap placement on the participants' head and recording in 19-channel (Fp1, Fp2, F7, F3, Fz, F4, F8, T3, T4, T5, T6, C3, Cz, C4, P3, Pz, P4, O1 and O2). The EEG signals data were recorded in .EDF file format and converted into .EEG file format using Mitsar® EEG Studio Processing Software for spectral analysis using WinEEG software version 2.90.

The sampling rate was set at 1 kHz, high pass filter at 0.3 Hz, low pass filter at 50 Hz, notch filter at 45–55 Hz, and the recording was carried out in average reference montage. The band pass filter was set at a range of 0.3 Hz to 49 Hz to remove the low and high frequency noises and other artifacts such as electrical power line interference, eye movement, and muscles activities, from the raw EEG data. As all subjects were working doctors without medical history, we do not expect any potential seizures, and since gamma waves are not the focus of the study in sleepiness and fatigue, setting low pass filter at 50 Hz is acceptable. The notch filter in WinEEG software is set as a range and since the electrical frequency of Malaysia is 50 Hz, the notch filter is set as 45–55 Hz. The average montage was preferred during analysis to identify the brain waves that stand out in background significantly for the changes in spectral power before and after on-call sessions.

During the recording, the participants were seated in front of a laptop placed on a writing table with a black dot on its screen. The first EEG recording was done in the morning before 10 am on the day of the on-call duty, and the second was done in the afternoon after 3.30 pm on the post-call day. Ideally the EEG should be performed before the start and after the end of the on-call period, namely 8 am and 5 pm respectively. However, due to logistic reasons, where the doctors were required to conduct rounds and wrap up before the start and end of the o-call period, we conducted EEG later than 8 am but set to complete it before 10 am for the before on-call data collection, and between 3.30 pm to 5 pm for after on-call data collection. The EEG recording was carried out continuously for 14 minutes, starting with 3.5 minutes of open eye, followed by 3.5 minute of close eye, then 3.5 minute of re-open eye, and finally 3.5 minute of completing the Stroop Test. The research assistant timed the recording and instructed verbally the participant to close eye, open eye and perform Stroop Test at the stipulated time.

The EEG signal was computed using a short time Fourier transformation every second with 50% overlap by the Welch periodograms method. The output power spectral density was measured in microvolt square ($\mu V^2$) corresponding to the respective brain wave defined by its typical frequency range, as per default setting in the WinEEG software, namely theta (4 to 7.5 Hz), alpha (7.5 to 15 Hz) and beta (above 14 Hz). We also calculated three additional derived relative theta, alpha and beta values, by obtaining the ratio of the respective power spectral density value of the corresponding brain waves over the total EEG spectral power. We compared the power spectral density and the derived relative values across different brain regions namely frontal (Fp1, Fp2, F7, F3, Fz, F4, F8), centro-parietal (C3, Cz, C4, P3, Pz, P4), temporal (T3, T4, T5, T6), occipital (O1, O2) and globally, before and after the overnight on-call duty. The differences in the values before and after the overnight on-call duty were tested by paired t-test using SPSS version 20.

We carried out the Stroop Test during the last section of EEG monitoring to detect any deterioration in the participants' attention capacity and processing speed following a night of on-call duty. The test required the participants to press the key of the first letter of the

displayed word's colour. For an example, if the word 'BLUE' is displayed in red colour, the participant will have to press the 'B' key on the keyboard. There were six colours (red, yellow, blue, green, purple, and orange) with their corresponding word used in this test. The colours and corresponding words were presented either congruent or incongruently in a random manner for each trial. After each trial, the participant will have to hit the space bar to proceed to the next trial. A total of 21 trials were conducted, and the speed (in milliseconds) to answer each trial and the percentage correct were recorded.

The study was registered with the Malaysian National Medical Research Registry (NMRR) (ID: NMRR-18-3104-44693) and obtained ethical approval from the Malaysian Research Ethics Committee (MREC) (KMM/NIHSEC/P19-90(12)) and Universiti Malaysia Sarawak Medical Research Ethics Committee (UNIMAS/NC-21.02t03-02 Jld.3 (74)). Written informed consent were obtained from all participants before the commencement of the study.

## Results

A total of 32 volunteer doctors consented to participate in this study. The volunteered participants consisted of 24 male and 8 female doctors, with then mean (SD; min; max) age of 30 (2; 27; 35) years. All participants declared no history of chronic medical illness, and alcohol consumption or smoking 24 hours prior to the study. There were 24 participants declared drinking some amount of tea or coffee during the on-call period. All participants had basic medical degree, and two of them had additional postgraduate degree. The mean (SD; min; max) duration of working was 5 (2; 2; 11) years, and the body mass index was 25.8 (5.2; 17.2; 37.2) kg/m$^2$.

Prior to the on-call duty, the participants declared mean (SD; min; max) usual overnight sleep duration of 6.7 (1.3; 5.0; 10) hours. The mean (SD; min; max) sleep duration of all participants during the on-call period was 2.2 (1.2; 0.5; 5.0) hours (p<0.001, paired t-test), significantly shorter than their usual overnight sleep duration. The mean (SD) Chalder Fatigue Scale score of the participants were 10.8 (5.3) before on-call and 18.4 (6.6) after on-call (p-value < 0.001). The mean scores for individual items in the scale were all less than 1.3 before the on-call but more than 1.6 after the on-call. This indicates all participants responded either tired, fatigued, or sleepy after the overnight on-call. Table 1 shows the distribution of participants according to Chalder Fatigue Scale cut-off score at 15. Seventy eight percent of the after on-call doctors scored 'fatigue' and 84% of before on-call doctors scored 'no fatigue' using Chalder Fatigue Scale. The difference was statistically significant using McNemar's test, indicating the doctors after on-call were indeed in the 'fatigue' status. With this verification, it is meaningful to compare the power spectral differences in the EEG rhythms before and after on-call among the doctors.

The average power spectrum comparison of brain waves at different brain regions for all participants before and after the on-call duty is shown in Table 2. The alpha rhythm attenuation during eye open was consistent across all brain regions and the theta rhythm was maximal at the frontal regions in both sessions (before and after on-call). These are consistent with

**Table 1. Distribution of participants based on Chalder Fatigue Scale cut-off of 15 before and after an overnight on-call.**

| Chalder Fatigue Scale score | <15 | ≥ 15 |
|---|---|---|
| Before on-call, n (%) | 27 (84.4) | 5 (15.6) |
| After on-call, n (%) | 7 (21.9) | 25 (78.1) |

Notes: McNemar's test p-value < 0.001

**Table 2. Power spectrum comparison of electroencephalogram before and after on-call duty for all participants.**

| Brain waves (band range, Hz) | Phase | Global | | | Frontal | | | Centro-parietal | | | Temporal | | | Occipital | | |
|---|---|---|---|---|---|---|---|---|---|---|---|---|---|---|---|---|
| | | Pre-call | Post-call | p-value | Pre-call | Post-call | p-value | Pre-call | Post-call | p-value | Pre-call | Post-call | p-value | Pre-call | Post-call | p-value |
| **Theta (1.5–4)** | Open eye | 1.902 | 2.107 | 0.630 | 1.327 | 1.239 | 0.842 | 0.369 | 0.380 | 0.855 | 0.518 | 0.426 | 0.373 | 0.213 | 0.233 | 0.779 |
| | Close eye | **1.561** | **2.060** | **0.036** | 0.610 | 1.048 | 0.074 | **0.395** | **0.582** | **0.039** | 0.504 | 0.483 | 0.777 | 0.247 | 0.331 | 0.229 |
| | Reopen eye | 2.306 | 2.311 | 0.994 | 1.381 | 1.240 | 0.758 | 0.417 | 0.491 | 0.494 | 0.503 | 0.466 | 0.737 | 0.257 | 0.310 | 0.660 |
| | Stroop test | 4.121 | 3.836 | 0.739 | 2.239 | 1.966 | 0.643 | 0.709 | 0.447 | 0.099 | 0.819 | 0.611 | 0.088 | 0.354 | 0.252 | 0.092 |
| **Alpha (4–7.5)** | Open eye | 3.782 | 3.573 | 0.796 | 1.388 | 1.338 | 0.890 | 1.033 | 0.857 | 0.327 | 1.353 | 0.958 | 0.109 | 0.750 | 0.686 | 0.792 |
| | Close eye | 6.581 | 4.720 | 0.081 | 1.580 | 1.862 | 0.595 | 1.431 | 1.173 | 0.367 | **2.276** | **1.366** | **0.007** | 1.525 | 1.108 | 0.157 |
| | Reopen eye | 3.066 | 3.166 | 0.858 | 1.068 | 1.484 | 0.305 | 0.738 | 0.915 | 0.304 | 1.017 | 0.744 | 0.156 | 0.500 | 0.538 | 0.763 |
| | Stroop test | 3.206 | 2.631 | 0.306 | 1.336 | 1.076 | 0.242 | 0.649 | 0.587 | 0.740 | 0.861 | 0.829 | 0.863 | 0.358 | 0.323 | 0.621 |
| **Beta (7.5–14)** | Open eye | 1.297 | 1.170 | 0.456 | 0.792 | 0.824 | 0.906 | 0.279 | 0.237 | 0.382 | **0.434** | **0.278** | **0.043** | 0.132 | 0.096 | 0.205 |
| | Close eye | 1.137 | 0.985 | 0.188 | 0.410 | 0.553 | 0.297 | 0.304 | 0.255 | 0.211 | **0.381** | **0.249** | **0.010** | 0.167 | 0.118 | 0.237 |
| | Reopen eye | 1.469 | 1.157 | 0.200 | 0.747 | 1.043 | 0.465 | 0.284 | 0.263 | 0.753 | 0.429 | 0.232 | 0.059 | 0.147 | 0.093 | 0.208 |
| | Stroop test | 1.718 | 2.027 | 0.506 | 0.736 | 0.728 | 0.953 | 0.337 | 0.292 | 0.608 | 0.500 | 0.461 | 0.645 | 0.143 | 0.127 | 0.357 |
| **Relative Theta** | Open eye | 0.287 | 0.373 | 0.167 | 0.334 | 0.404 | 0.073 | 0.262 | 0.316 | 0.087 | 0.243 | 0.291 | 0.120 | 0.245 | 0.296 | 0.161 |
| | Close eye | **0.210** | **0.269** | **0.029** | **0.271** | **0.343** | **0.001** | **0.215** | **0.312** | **0.000** | **0.182** | **0.263** | **0.001** | **0.159** | **0.257** | **0.001** |
| | Reopen eye | 0.304 | 0.339 | 0.297 | 0.355 | 0.404 | 0.166 | **0.288** | **0.349** | **0.033** | **0.261** | **0.326** | **0.040** | 0.265 | 0.330 | 0.060 |
| | Stroop test | 0.426 | 0.423 | 0.924 | 0.469 | 0.444 | 0.497 | 0.401 | 0.370 | 0.239 | 0.365 | 0.346 | 0.449 | 0.395 | 0.397 | 0.969 |
| **Relative Alpha** | Open eye | 0.490 | 0.549 | 0.621 | 0.410 | 0.375 | 0.298 | 0.505 | 0.505 | 0.982 | 0.523 | 0.495 | 0.486 | 0.581 | 0.566 | 0.729 |
| | Close eye | **0.617** | **0.502** | **0.000** | **0.542** | **0.486** | **0.031** | 0.592 | 0.534 | 0.068 | 0.649 | 0.585 | 0.052 | **0.714** | **0.637** | **0.026** |
| | Reopen eye | 0.440 | 0.427 | 0.608 | 0.366 | 0.367 | 0.957 | 0.464 | 0.483 | 0.548 | 0.499 | 0.495 | 0.901 | 0.535 | 0.544 | 0.820 |
| | Stroop test | 0.355 | 0.333 | 0.244 | 0.324 | 0.317 | 0.731 | 0.374 | 0.400 | 0.214 | 0.392 | 0.403 | 0.546 | 0.407 | 0.412 | 0.829 |
| **Relative Beta** | Open eye | 0.223 | 0.275 | 0.324 | 0.255 | 0.221 | 0.237 | 0.232 | 0.180 | 0.019 | 0.234 | 0.214 | 0.513 | 0.174 | 0.138 | 0.004 |
| | Close eye | 0.173 | 0.298 | 0.127 | 0.187 | 0.171 | 0.161 | **0.193** | **0.154** | **0.024** | 0.170 | 0.152 | 0.231 | 0.127 | 0.106 | 0.078 |
| | Reopen eye | 0.256 | 0.468 | 0.392 | 0.279 | 0.228 | 0.084 | **0.248** | **0.168** | **0.001** | **0.241** | **0.179** | **0.005** | **0.200** | **0.126** | **0.000** |
| | Stroop test | 0.218 | 0.243 | 0.239 | 0.207 | 0.238 | 0.225 | 0.226 | 0.230 | 0.816 | 0.243 | 0.251 | 0.720 | 0.198 | 0.191 | 0.600 |

Notes:

1. Bold values indicate statistically significant changes in power density after overnight on-call duty.

2. Values represent average power spectral density in microvolts square, $\mu V^2$, for all participants.

3. P-values were obtained by paired t-test.

normal adult EEG patterns. The theta rhythm showed significant increase in power globally after an overnight on-call duty, especially when measured in eye closure phase. In contrast, the alpha and beta rhythms showed reduction in power globally, significantly at temporal region following an overnight on-call duty, when measured in eye closure phase. These effects are more statistically significant when we calculated the respective relative values, namely relative theta, relative alpha, and relative beta as shown in the Table 2.

**Table 3. The Stroop test result during EEG recording before and after an overnight on-call.**

| Stroop test | Before on-call | After on-call | p-value |
|---|---|---|---|
| Number of trials attempted | 99.9 (16.6) | 105.0 (16.3) | 0.121 |
| Response time (msec) | | | |
| Congruent trials | 1031.8 (257.2) | 1068.3 (280.5) | 0.554 |
| Incongruent trials | 1312.6 (332.8) | 1329.8 (361.1) | 0.844 |
| Correct percentage (%) | | | |
| Congruent trials | 99.6 (1.3) | 98.6 (2.5) | 0.036 |
| Incongruent trials | 97.6 (2.9) | 92.5 (14.6) | 0.055 |

Notes: Values in the cells represent mean (SD). P-values were obtained using paired t-test.

The Stroop test alone showed despite answering more trials, there were longer response time and lower correct percentage for both congruent and incongruent trials among the participants after on-call as compared to before on-call, although the differences were not statistically significant (Table 3). This is consistent with the reduced cognitive capability after an overnight on-call duty. Nevertheless, the EEG power spectrum during Stroop test did not show significant differences before and after on-call duty.

## Discussion

A normal adult exhibit small amplitude beta rhythm during awake, posterior dominant alpha rhythm during relaxation, and appearance of theta rhythm from drowsiness to sleep [27]. In our study, we found significant increase in the spectral power of theta rhythm after an overnight on-call sleep deprivation, especially when we measured the EEG signal during eye closure. Conversely, both alpha and beta rhythms showed reduction in spectral power at post-call compared to pre-call session. These observations were more apparent when we calculated the respective relative values against the total EEG spectral power. The findings were consistent with the cognitive assessment responses—indicating the participants were experiencing mental fatigue following an overnight on-call duty with average sleeping duration of two hours—and the result of Stroop test.

The findings of this study are in consistence with studies conducted among long distance drivers that showed an increase in theta activity associated with driver's fatigue which is likely to reflect the decrease in cortical arousal [17, 28–30]. Studies investigating driver's fatigue utilised continuous monitoring of EEG over a period of time, which is impractical if it is going to be implemented among the doctors during on-call duty. For on-call doctors, it is of utmost important to detect mental fatigue or sleepiness following an overnight on-call, to prevent them from performing further duty which may endanger themselves or patients due to the nature of the duty.

Cognitive assessment questionnaires and tests could be used as single time point screening tools for mental fatigue detection [29, 31–33]. However, these tools are subjective, time consuming and subject to respondent and measurement biases. On the other hand, EEG is more objective when it is carried out in a standard protocol. Based on the results of this study, it is possible to detect increase and decrease in the spectral power of the theta rhythm and alpha and beta rhythms respectively, at two time points, before and after the on-call duty. Hence, the finding of this study is useful for possible development a simplified EEG screening tool with algorithms to detect mental fatigue or sleepiness for workers performing high risk job.

Despite the presence of spectral power differences between the before and after on-call recordings, there was lack of statistical significance when the results were stratified according to brain regions. This could be due to small number of samples. Besides, the study was not carried out under a controlled induced mental fatigue environment. There was a dilution in the after on-call data as not all participants were deprived of sleep in this study. The inconsistency in the nature of the on-call duty among the participants and undergoing of regular on-calls among the participants prior to the conduct of the study could have further confounded the results of this study. Despite these limitations, the current study revealed the differences in the EEG signals among the doctors before and after an overnight on-call duty in a real-life situation.

In the current study, we used Chalder Fatigue Scale to elucidate self-perceived fatigue among the study subjects. The result presented in Table 1 supports the presence of self-perceived fatigue among the doctors after on-call. We did not use other questionnaires due to time constraints. From the physiological point of view, it is intuitive—from the fact that the mean sleeping duration for on-call doctors in our study was only 2.2 hours—that the study subjects should be in the state of sleep deprived condition after on-call.

We acknowledge the limitation of small sample size in this study, as it was extremely difficult to get full cooperation of on-call doctors to undergo the EEG test before and after their hectic on-call period. The proposed model of EEG response to sleep deprivation with possible fatigue may not be confidently generalisable to a bigger population but the fact that the differences in theta waves globally and at centro-parietal regions were statistically significant despite small differences and sample size warrants attention. We recommend similar studies to be conducted in the future so that systematic reviews and meta-analyses could be carried out to consolidate the evidence.

## Conclusion

Following an overnight on-call duty, the spectral power of the theta rhythm increases globally, and the alpha and beta rhythm decrease mainly at temporal region, most prominent during eye closure measurement. These changes were observed after a period of sleep deprived on-call duty with average of only 2.2 hours of sleep and were consistent with the average Chalder Fatigue Scale score of 18.4.

## Supporting information

**S1 Data.**
(CSV)

## Author Contributions

**Conceptualization:** Anselm Ting Su.

**Data curation:** Anselm Ting Su, Gregory Xavier.

**Formal analysis:** Anselm Ting Su, Gregory Xavier.

**Investigation:** Anselm Ting Su, Gregory Xavier, Jew Win Kuan.

**Methodology:** Anselm Ting Su, Gregory Xavier, Jew Win Kuan.

**Project administration:** Anselm Ting Su.

**Supervision:** Anselm Ting Su, Jew Win Kuan.

**Writing – original draft:** Anselm Ting Su.

**Writing – review & editing:** Anselm Ting Su, Jew Win Kuan.

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
