## [Decision Letter · Decision Letter 0]

11 Nov 2022

PONE-D-22-25035The measurement of mental fatigue following an overnight on-call duty among doctors using electroencephalogramPLOS ONE

Dear Dr. Su,

Thank you for submitting your manuscript to PLOS ONE. After careful consideration, we feel that it has merit but does not fully meet PLOS ONE’s publication criteria as it currently stands. Therefore, we invite you to submit a revised version of the manuscript that addresses the points raised during the review process.

We look forward to receiving your revised manuscript.

Kind regards,

Jerritta Selvaraj

Academic Editor

PLOS ONE

Journal Requirements:

“This study was funded by the Brain, Mind and Neuroscience Research Foundation Malaysia (Registration: PPAB-26/2017) with research grant ID: GL/F05/YPOMNM/2019.”

“This study was funded by the Brain, Mind and Neuroscience Research Foundation Malaysia (Registration: PPAB-26/2017) (URL: https://www.ypomnm.org) with research grant ID: GL/F05/YPOMNM/2019. The recipient of the grant was AST. he funders had no role in study design, data collection and analysis, decision to publish, or preparation of the manuscript.”

Reviewers' comments:

Reviewer's Responses to Questions

**Comments to the Author**

1. Is the manuscript technically sound, and do the data support the conclusions?

Reviewer #1: Partly

Reviewer #2: Partly

2. Has the statistical analysis been performed appropriately and rigorously? 

Reviewer #1: Yes

Reviewer #2: Yes

3. Have the authors made all data underlying the findings in their manuscript fully available?

Reviewer #1: No

Reviewer #2: Yes

4. Is the manuscript presented in an intelligible fashion and written in standard English?

Reviewer #1: Yes

Reviewer #2: Yes

5. Review Comments to the Author

Reviewer #1: 1. It is recommended to add a detailed description of contributions in the introduction to strengthen motivation.

2. A major problem in this study is that it does not even has any objectives (or hypotheses). So, how can the methodological approach be explained?

3. In the Abstract, it is written that some statistical techniques, like Chalder Fatigue Scale score and SPSS were used to analyse some data, but, where are the results from this analyses detailed in the paper; and what are the objectives that the results obtained from these analyses were meant to verify

4. The author has informed that 19 channels has been used to obtain EEG data, But in results the author has mentioned as Global, Frontal, Centro-parietal, Temporal and Occipital regions. What are the 19 nodes used in this study and how the author has combined the result in region wise (Global, Frontal, Centro-parietal, Temporal and Occipital regions).

5. What is the reason for choosing alpha and beta bands in this study. Any references

6. Whether any artifacts present in the EEG data? How the artifacts are removed. Since I didn’t see any filtering techniques in the article.

7. The author can add some images related to EEG data, Filtered data for better understanding of data analysis

8. This study mainly focuses on Chalder Fatigue Scale score. Whether the CFSS predict the fatigue state accurately, since it is a questionnaire?

Reviewer #2: The sampling rate was set at 1 kHz, low pass filter at 0.3 Hz, high pass filter at 50 Hz, notch filter at 45 – 55 Hz, and the recording was carried out in average reference montage. Justify the selection of these values.

The first EEG recording was done in the morning before 10 am on the day of the on-90 call duty, and the second was done in the afternoon after 3.30 pm on the post-call day. Justify the selection of these timings.

Sample size is tool small - A total of 32 volunteer doctors – Justify the accuracy of the proposed model.

The study relies only on only EEG characteristics. Is there any other methods to detect resting state drowsiness or mental fatigue.

The authors are advised to include more data to improve the quality of the paper.

6. PLOS authors have the option to publish the peer review history of their article (what does this mean?). If published, this will include your full peer review and any attached files.

Reviewer #1: **Yes: **Dr. S. PRADEEP KUMAR

Reviewer #2: No

---

## [Author Response · Author response to Decision Letter 0]

23 Feb 2023

We would like to response to the reviewers’ comments as follows:

Reviewer #1: 

1. It is recommended to add a detailed description of contributions in the introduction to strengthen motivation.

Response: We have added in statements on the contributions of the study at the end of the Introduction section.

2. A major problem in this study is that it does not even has any objectives (or hypotheses). So, how can the methodological approach be explained?

Response: We have added in the objective statement clearly after the problem statement in the Introduction section.

3. In the Abstract, it is written that some statistical techniques, like Chalder Fatigue Scale score and SPSS were used to analyse some data, but, where are the results from this analyses detailed in the paper; and what are the objectives that the results obtained from these analyses were meant to verify.

Response: The results of Chalder Fatigue Scale questionnaire was presented in the second paragraph of the Results section and Table 1. We used the cut-off point of 15 to differentiate subjects identified as fatigue versus not fatigue during both pre-call and post-call situations based on a previous study by Wessely et al. (1997) (Wessely, S., Chalder, T., Hirsch, S., Wallace, P., & Wright, D. (1997). The prevalence and morbidity of chronic fatigue and chronic fatigue syndrome: a prospective primary care study. American Journal of Public Health, 87(9), 1449-1455.). Table 1 is meant to show that the post-call doctors were indeed ‘fatigued’ based on their response to the Chalder Fatigue Scale questionnaire. The spectral power differences in the EEG rhythms reported later in the manuscript hence correspond to the fatigue status of the pre-call and post-call doctors respectively. We have added in a reference and descriptions to justify the cut-off point that we used for Chalder Fatigue Scale in the Materials and Methods section and explained the implications of Table 1 result in the Results section.

4. The author has informed that 19 channels has been used to obtain EEG data, But in results the author has mentioned as Global, Frontal, Centro-parietal, Temporal and Occipital regions. What are the 19 nodes used in this study and how the author has combined the result in region wise (Global, Frontal, Centro-parietal, Temporal and Occipital regions).

Response: The 19-channel EEG of 10/20 system is standard, namely (Fp1, Fp2, F7, F3, Fz, F4, F8, T3, T4, T5, T6, C3, Cz, C4, P3, Pz, P4, O1 and O2). The respective brain regions are: frontal (Fp1, Fp2, F7, F3, Fz, F4, F8), centro-parietal (C3, Cz, C4, P3, Pz, P4), temporal (T3, T4, T5, T6), and occipital (O1, O2). We have added in the channel indicators for respective brain regions in the Materials and Methods section.

5. What is the reason for choosing alpha and beta bands in this study. Any references.

Response: We did not specifically choose alpha and beta waves in this study. We analysed alpha, beta and theta waves as these are normally present in an awake person and compared the power spectral of those waves before and after on-call session. We did not choose delta waves as it is considered abnormal in awake subjects. From the analysis, we found that theta waves increased in after on-call situations which is consistent with the existing knowledge that theta waves represent drowsiness.

6. Whether any artifacts present in the EEG data? How the artifacts are removed. Since I didn’t see any filtering techniques in the article.

Response: The artifacts were removed by filtering mechanism reported in the third paragraph of Materials and Methods data.

7. The author can add some images related to EEG data, Filtered data for better understanding of data analysis.

Response: The EEG tracing before and after application of filter is shown below. The EEG tracing is smoother and provides better visualisation of brain waves after filtering. As there are 64 continuous 14 minutes tracings of EEG signals, the authors feel that it is not appropriate to include images of any snapshot of EEG data showing before and after filter application in this manuscript. The EEG test was carried out under standard protocol of a clinical EEG test setting including precautions to remove artifacts and filter setting. We have added in additional descriptions on this matter in the Materials and Methods section.

Image before filtering: (Image is available in Word file "Response to Reviewers")

Image after filtering: (Image is available in Word file "Response to Reviewers")

8. This study mainly focuses on Chalder Fatigue Scale score. Whether the CFSS predict the fatigue state accurately, since it is a questionnaire?

Response: As per the objective of this study, the study focuses on comparing the power spectral of brain waves between before and after on-call doctors. It is not focused on Chalder Fatigue Scale, the Chalder Fatigue Scale is a tool to provide the likelihood of the after on-call doctors experiencing fatigue based on their subjective response as compared to before on-call. We have added additional statements in the Results section regarding this.

Reviewer #2: 

1.The sampling rate was set at 1 kHz, low pass filter at 0.3 Hz, high pass filter at 50 Hz, notch filter at 45 – 55 Hz, and the recording was carried out in average reference montage. Justify the selection of these values.

Response: Sorry, there was a typo error. The 0.3 Hz is referring to high pass filter (or low cut filter) and the 50 Hz is referring to low pass filter (or high cut filter). The sampling rate of 1 kHz is a common standard in our clinical EEG test in hospital setting. The high pass filter is set at 0.3 Hz to remove low-frequency noise and artifacts, while preserving the slow wave components of the EEG signal, such as the delta waves in deep sleep. A higher cut-off frequency for the high pass filter could potentially remove important slow wave components of the EEG signal. (Please take note that there is no 0.5 filter in WinEEG software, and hence we selected the 0.3 Hz, which we think is acceptable). The low pass filter is set at 50 Hz to remove high-frequency noise and artifacts, while preserving the majority of the EEG signal. As all subjects were working doctors without medical history, we do not expect any potential seizures and we are not interested in gamma waves, hence setting low pass filter at 50 Hz is acceptable. The notch filter in WinEEG software is set as a range and since the electrical frequency of Malaysia is 50 Hz, the notch filter is set as 45 – 55 Hz. The average montage was used during analysis because we want to identify which brain waves stand out in background significantly for the changes in spectral power before and after on-call sessions. We have added the above explanation in the Materials and Methods section and made the corrections accordingly.

2. The first EEG recording was done in the morning before 10 am on the day of the on-90 call duty, and the second was done in the afternoon after 3.30 pm on the post-call day. Justify the selection of these timings.

Response: The justification of the timings is mainly logistic. An on-call doctor in our samples typically worked from 8 am to 5pm the next day. The doctor was not expected to sleep at night but was encouraged to take short nap whenever the clinical work is permissible. Ideally the EEG should be performed before the start and after the end of the on-call period. However, due to practical reasons, where the doctors were required to conduct rounds and wrap up before the start and end of the o-call period, we conducted EEG later than 8 am but set to complete it before 10 am for the before on-call data, and between 3.30 pm to 5 pm for after on-call data. We have added the justification in the Materials and Methods section. 

3. Sample size is tool small - A total of 32 volunteer doctors – Justify the accuracy of the proposed model.

Response: We acknowledge the limitation of small sample size and it was extremely difficult to get full cooperation of on-call doctors to undergo the EEG test before and after their hectic on-call period. The proposed model of EEG response to sleep deprivation with possible fatigue may not be confidently generalisable to bigger population but the fact that the values are statistically significant despite small differences and sample size warrants attention. We recommend further similar studies to be conducted so that future systematic reviews and meta-analysis could be carried out to consolidate the evidence. We have included this limitation and recommendation in the Discussion section.

4. The study relies only on only EEG characteristics. Is there any other methods to detect resting state drowsiness or mental fatigue.

Response: We used Chalder Fatigue Scale to elucidate self-perceived fatigue among the study subjects and the result in Table 1 supports the presence of self-perceived fatigue among the doctors after on-call. We did not use other questionnaires due to time constraints. It is intuitive from the fact that the mean sleeping duration for on-call doctors in our study was only 2.2 hours, that the study subjects should be in the state of sleep deprived condition after on-call, from the physiological point of view. We have added this statement in the Discussion section.

5. The authors are advised to include more data to improve the quality of the paper.

Response: Thank you for the advice and we totally agree with the recommendation. It is a difficult limitation to overcome when it comes to approaching doctors as the study subjects. Nevertheless, we hope to report the findings that we have so far as we think it is valuable to add to the current body of knowledge.

Thank you.

---

## [Decision Letter · Decision Letter 1]

19 Jun 2023

The measurement of mental fatigue following an overnight on-call duty among doctors using electroencephalogram

PONE-D-22-25035R1

Dear Dr. Su,

We’re pleased to inform you that your manuscript has been judged scientifically suitable for publication and will be formally accepted for publication once it meets all outstanding technical requirements.

Kind regards,

Dragan Hrncic

Academic Editor

PLOS ONE

Additional Editor Comments (optional):

Reviewers' comments:

Reviewer's Responses to Questions

**Comments to the Author**

1. If the authors have adequately addressed your comments raised in a previous round of review and you feel that this manuscript is now acceptable for publication, you may indicate that here to bypass the “Comments to the Author” section, enter your conflict of interest statement in the “Confidential to Editor” section, and submit your "Accept" recommendation.

Reviewer #1: All comments have been addressed

2. Is the manuscript technically sound, and do the data support the conclusions?

Reviewer #1: Yes

3. Has the statistical analysis been performed appropriately and rigorously? 

Reviewer #1: Yes

4. Have the authors made all data underlying the findings in their manuscript fully available?

Reviewer #1: Yes

5. Is the manuscript presented in an intelligible fashion and written in standard English?

Reviewer #1: Yes

6. Review Comments to the Author

Reviewer #1: 1. What is the future scope of this work

2. The author can add some data collection images

3. Since the work related to mental fatigue was done by some authors, A comparison can be made

7. PLOS authors have the option to publish the peer review history of their article (what does this mean?). If published, this will include your full peer review and any attached files.

Reviewer #1: **Yes: **S. Pradeep Kumar

---

## [Editor Report · Acceptance letter]

22 Jun 2023

PONE-D-22-25035R1 

The measurement of mental fatigue following an overnight on-call duty among doctors using electroencephalogram 

Dear Dr. Su:

I'm pleased to inform you that your manuscript has been deemed suitable for publication in PLOS ONE. Congratulations! Your manuscript is now with our production department. 

Kind regards, 

on behalf of

Professor Dragan Hrncic 

Academic Editor

PLOS ONE